# A Putative D-Arabinono-1,4-lactone Oxidase, MoAlo1, Is Required for Fungal Growth, Conidiogenesis, and Pathogenicity in *Magnaporthe oryzae*

**DOI:** 10.3390/jof8010072

**Published:** 2022-01-11

**Authors:** Ming-Hua Wu, Lu-Yao Huang, Li-Xiao Sun, Hui Qian, Yun-Yun Wei, Shuang Liang, Xue-Ming Zhu, Lin Li, Jian-Ping Lu, Fu-Cheng Lin, Xiao-Hong Liu

**Affiliations:** 1State Key Laboratory for Managing Biotic and Chemical Threats to the Quality and Safety of Agro-Products, Institute of Biotechnology, Zhejiang University, Hangzhou 310058, China; wuminghuaecho@163.com (M.-H.W.); lhuang4@uni-koeln.de (L.-Y.H.); 21816077@zju.edu.cn (L.-X.S.); huiqian@zju.edu.cn (H.Q.); yy21716074@163.com (Y.-Y.W.); fuchenglin@zju.edu.cn (F.-C.L.); 2Biocenter, Institute for Plant Sciences, University of Cologne, 50674 Cologne, Germany; 3State Key Laboratory for Managing Biotic and Chemical Threats to the Quality and Safety of Agro-Products, Central Laboratory, Zhejiang Academy of Agricultural Sciences, Hangzhou 310021, China; liangs@zaas.ac.cn; 4State Key Laboratory for Managing Biotic and Chemical Threats to the Quality and Safety of Agro-Products, Institute of Plant Protection and Microbiology, Zhejiang Academy of Agricultural Sciences, Hangzhou 310021, China; zhuxm@zaas.ac.cn (X.-M.Z.); 11816015@zju.edu.cn (L.L.); 5College of Life Science, Zhejiang University, Hangzhou 310058, China; jplu@zju.edu.cn

**Keywords:** D-erythroascorbic acid, D-arabinono-1,4-lactone oxidase, *Magnaporthe oryzae*

## Abstract

*Magnaporthe oryzae* is the causal agent of rice blast outbreaks. L-ascorbic acid (ASC) is a famous antioxidant found in nature. However, while ASC is rare or absent in fungi, a five-carbon analog, D-erythroascorbic acid (EASC), seems to appear to be a substitute for ASC. Although the antioxidant function of ASC has been widely described, the specific properties and physiological functions of EASC remain poorly understood. In this study, we identified a D-arabinono-1,4-lactone oxidase (ALO) domain-containing protein, MoAlo1, and found that MoAlo1 was localized to mitochondria. Disruption of *MoALO1* (Δ*Moalo1*) exhibited defects in vegetative growth as well as conidiogenesis. The Δ*Moalo1* mutant was found to be more sensitive to exogenous H_2_O_2_. Additionally, the pathogenicity of conidia in the Δ*Moalo1* null mutant was reduced deeply in rice, and defective penetration of appressorium-like structures (ALS) formed by the hyphal tips was also observed in the Δ*Moalo1* null mutant. When exogenous EASC was added to the conidial suspension, the defective pathogenicity of the Δ*Moalo1* mutant was restored. Collectively, MoAlo1 is essential for growth, conidiogenesis, and pathogenicity in *M. oryzae.*

## 1. Introduction

*Magnaporthe oryzae* is a pathogen that threatens all crops around the world. It is the causal agent of rice blast outbreaks, which cause economically significant crop losses annually [1]. *M. oryzae* is a filamentous phytopathogenic fungus that has a complex life cycle with distinct developmental stages. The pathogen initiates as a three-celled asexual conidia that then forms a structure called an appressorium [2]. The cuticle and cell wall of plants are mechanically destroyed by the appressorium through the accumulation of up to 8 × 10^6^ Pa of turgor pressure [3]. Infected hyphae expand in plant cells, causing disease, and then form new conidia. In the form of conidia, *M. oryzae* disperses in paddy fields to start the second round of the infection cycle [4,5].

L-ascorbic acid (ASC) is one of the most famous biological antioxidants and has been found to participate in complex signaling pathways as well as mediate responses to biotic or abiotic stresses [6,7]. Increasing data indicate that ASC exists in plants and some animals, except for humans, other primates, guinea pigs, and some birds [8,9,10]. However, the synthesis of ASC varies in different organisms. Enzymes catalyzing the last step of ASC synthesis can be classified into three classes according to their substrate specificity, electron receptor, and subcellular localization [11]. In mammals, gulonolactone oxidase (GLO) recruits oxygen as an electron acceptor to catalyze ASC synthesis, and it is located in the endoplasmic reticulum [12,13]. In plants, L-galactoside-γ-lactone, which is located in the mitochondria, catalyzes the synthesis of ASC in the presence of the electron acceptor cytochrome c [7,14]. However, ASC does not seem to exist naturally in lower eukaryotes such as fungi [6,11,15,16], but a five-carbon analog, D-erythroascorbic acid (EASC), which shows a high degree of physicochemical and structural similarity to ASC, has been found to be a substitute for ASC [11].

The antioxidant function of EASC in *Saccharomyces cerevisiae* has been studied by predecessors [15]. Some evidence also suggests an association between fungi growth and free EASC. The accumulation of EASC was detected at a high level during the exponential phase of mycelial growth, but appeared to show a lower level during the stationary phase [16]. In addition, EASC plays important roles in the morphological regulation and selective respiration of cells [17]. However, the specific function of D-arabinono-1,4-lactone oxidase has not been fully elucidated in fungi yet. Previous studies showed that a deficiency in *ALO* increased sensitivity towards oxidative stress [15,18] and abnormal filamentation [18,19], stunted growth, and decreased virulence [16]. Therefore, a lack of *ALO* may cause the dysfunction of normal physiological and biochemical pathways. This may indicate that Alo might be involved in plant pathogenesis and fungal development.

In this study, we identified an ALO domain-containing protein from *M. oryzae*, MoAlo1, involved in EASC synthesis. To uncover the function of MoAlo1, we disrupted *MoALO1* in *M. oryzae* and analyzed the resulting phenotypes. We found that MoAlo1 exerts its function on fungal developmental processes of the rice blast fungus and that the Δ*Moalo1* mutant showed impaired fungal growth, conidiogenesis, and pathogenicity and plays a role in oxidative stress. Additionally, MoAlo1 was found to be localized to mitochondria.

## 2. Materials and Methods

### 2.1. Fungal Strains and Culture Conditions

All strains used in this study (including the wild-type Guy11 and Δ*Moalo1* mutant, the complementary strain MoAlo1C, Δ*Moalo1*::ALO, and Δ*Moalo1*::FAD) were cultured in complete medium (CM), Yeast glucose (YG) medium, V8 medium, and oatmeal agar (OMA) medium at 25 °C under a 16 h light/8 h dark photoperiod for 8 days according to a previous study [20,21].

### 2.2. Generation of the MoALO1 Null Mutant and Complementation Strain in M. oryzae

In order to further determine the role of *MoALO1* in the rice blast fungus, we constructed a gene-deletion cassette followed by a high-throughput gene knockout system [22]. The pKO1B vector was digested using the restriction enzymes *Hin*dIII and *Xba*I. Next, an approximately 1.5 kb upstream fragment and a downstream fragment of *MoALO1* in *M*. *oryzae* genomic DNA were amplified with the primers MoALO1 up (F)/MoALO1 up (R) and MoALO1 down (F)/MoALO1 down (R), respectively (Table A1 in Appendix B). Then, a bacterial bialophos resistance gene (*BAR*) was amplified from pBARKS1 with the primers BAR-F/BAR-R. During the experiment, all PCR products and linearized vectors were verified by agarose gel electrophoresis and then purified using a DNA gel extraction kit (Axygen, Hangzhou, China). Then, both the upstream and downstream fragments of *MoALO1* and the *BAR* fragment were fused to pKO1B to construct a gene-deletion cassette vector. Then, the gene-deletion cassette was transferred to Guy11 using the *Agrobacterium tumefaciens*-mediated transformation (ATMT) method. Transformants were selected on CM medium with 300 μg/mL glufosinate ammonium. Then, for further confirmation, the genomic DNAs of the transformants were isolated using an improved CTAB method. The genomic DNAs were screened by a double PCR protocol using the *β-tubulin* gene as a positive control as previously reported [22]. Since the housekeeping genes are stably expressed in cells, in order to verify whether the null mutant has only one copy of the gene-deletion cassette, we used the *β-tubulin* gene (only one copy in the *M*. *oryzae* genome) as a control, and the number of copies of *BAR* was determined by Real-Time Quantitative Reverse Transcription polymerase chain reaction (qRT-PCR) to determine whether there is only one copy of the gene-deletion cassette in the Δ*Moalo1* mutant. The relative expression analysis was performed by determining the Ct value of each gene and normalized by the Ct value of *β-tubulin*. The primers used are listed in Table A1. As for the complementary strains, the *MoALO1* fragment without the stop codon (TAA) was amplified in the genomic DNA of *M*. *oryzae* and fused with GFP at the *Bam*HI site of pKD5. The validated vector was transformed into the Δ*Moalo1* mutant using the ATMT method. Media supplemented with 100 μg/mL sulfonylurea were used for mutant screening. As for Δ*Moalo1*::ALO and Δ*Moalo1*::FDA, the ALO or FDA domain was amplified using the specific primers listed in Table A1 and transferred into the Δ*Moalo1* mutant through the ATMT method, respectively.

### 2.3. Cellular Localization of MoAlo1 in M. oryzae

For the subcellular localization analysis, conidia were harvested from 8-day-old CM agar plates and incubated with 100 nM MitoTracker^®^ Red CMXRos (Invitrogen, Carlsbad, CA, USA), a red fluorescent mitochondrial dye, for 25–45 min. After staining, the sample was washed with phosphate-buffered saline (PBS) and then observed through a confocal fluorescence microscope (Zeiss LSM 710, 63 × oil, CarlZeiss, Jena, Germany) [23].

### 2.4. The Growth, Conidiation, Glycogen Distribution, and Oxidative Stress of M. oryzae

In order to measure the mycelial growth and conidiation, a 5-mm agar block of 8-day-old wild-type Δ*Moalo1* mutant and the complementary strain MoAlo1C were inoculated in the center of CM agar plates at 25 °C with a 16 h light and 8 h dark phase, respectively. Eight days post inoculation (dpi), the growth diameter of the strains was measured and the strains were photographed. For the conidia production, conidia were harvested using sterile water and counted by a hemocytometer under a microscope as previously reported [21]. For the distribution of glycogen experiment, conidia were harvested and diluted to 5 × 10^4^ conidia/mL. Then, 20 μL of conidial suspension was dropped onto a plastic film to induce the appressorium, which was then cultivated in the dark for 0, 4, 8, and 24 h and observed under the microscope after being stained with KI/I_2_ for 1 min. For the oxidative stress experiment, YG medium was supplemented with 10 mM hydrogen peroxide (H_2_O_2_) and the strains were grown in the dark at 25 °C for 8 days before being analyzed and photographed. The growth inhibition rate = (the diameter of strain growth on the YG medium for 8 dpi − the diameter of strain growth on the YG medium supplementary with H_2_O_2_ for 8 dpi)/the diameter of strain growth on the YG medium for 8 dpi.

### 2.5. Pathogenicity and Penetration Assays

To further confirm the pathogenicity of the strains, we conducted pathogenicity experiments on two susceptible hosts, rice (*Oryza sativa* cv. CO-39) and barley (*Hordeum vulgare* cv. ZJ-8). In the barley pathogenicity test, mycelial blocks of the same size or 5 × 10^4^ conidia/mL conidial suspensions were inoculated on eight-day-old barley leaves and then placed in a humid box at 25 °C for 16 h under light and 8 h in darkness alternately for 4 days. Then, the pathogenicity of the strains was observed and photographed after 4 days of inoculation [21]. The conidial suspensions eluted from the CM medium were diluted to 1 × 10^5^, 5 × 10^4^, and 1 × 10^4^ conidia/mL using a hemocytometer, and then 20 μL of conidial suspension was dropped onto the rice seedlings and photographed at 5 dpi. For the rice spraying assay, 14-day-old rice seedlings were sprayed with a 5 × 10^4^ conidia/mL conidial suspension using an artist’s airbrush and 0.2% (*w*/*v*) gelatin as a control. The inoculated rice seedlings were first placed in a dew chamber at 25 °C for 48 h in the dark, then transferred to a humid growth chamber with a 12 h light phase for 5 days. The disease symptoms were observed and photographed at 7 dpi [21]. Photoshop (PS) software was used to calculate the disease score by calculating the area of the lesion. The disease score = the area of the diseased spot/the area of the total leaf. As for the pathogenicity test, after adding exogenous EASC, 25 mM EASC was added to a 1 × 10^4^ conidia/mL conidial suspension and 20 μL of conidial suspension was dropped onto the detached rice leaves and then placed in a humid box at 25 °C for 16 h under light and 8 h in darkness alternately for 4 days before being photographed. For the detection of the penetration of appressoria formed by conidia, 20 μL of conidial suspension (1 × 10^4^ conidia/mL) was dropped onto 7-day-old barley leaves, which were then placed in a humid box at 25 °C for 24 h [21]. For the detection of the penetration of appressorium-like structures (ALS) formed by the hyphal tips, mycelial plugs were inoculated on 7-day-old barley leaves and then placed in a humid box at 25 °C. At 1, 2, 4, and 6 dpi, the barley leaves were faded with methanol and fixed in alcoholic lactophenol. After these treatments, the leaves were observed and photographed under a microscope [24].

For the stability and accuracy of the experiment, each test was repeated at least three times, with five replicates each time. All statistical analyses in our study were performed based on a t test (*p* < 0.01).

## 3. Results

### 3.1. Identification and Subcellular Localization of MoAlo1

Pfam domain analysis was performed on the *M. oryzae* proteome. We used the integrated module of the Pfam domain to search the CLC Genomics Workbench (QIAGEN, Aarhus, Denmark) with default parameters. The Pfam database used in the analysis was version 27. *MGG_02689* was found to contain the ALO domain (PF04030) and the FAD_binding_4 (PF01565) domain. Protein Basic Local Alignment Search Tool (BlastP) analysis showed that *MGG_02689* is similar to *ScALO1* [15] and designated as *MoALO1*. Sequence analysis using SMART showed that the ALO domain and FAD_binding_4 domains are also present in *S. cerevisiae* [15], *C. albicans* [16], *Schizosaccharomyces pombe*, *Neurospora crassa*, *Fusarium oxysporum*, *Colletotrichum gloeosporioides*, *Ustilaginoidea virens*, *Brassica oleracea* [25], *Rattus norvegicus* [26], and *Mus musculus* (Figure 1A). ALO and FAD_binding_4 domains are conserved in filamentous fungi, plants, and mammals. Additionally, to verify the genetic relationship of MoAlo1, we constructed a phylogenetic tree based on the MoAlo1 amino acid sequences and those of enzymes with similar activity to these organisms. As shown in Figure 1B, MoAlo1 showed a similar evolutionary relationship to those enzymes with similar activity in fungi, plants, and mammals.

To explore the subcellular localization of MoAlo1 in *M*. *oryzae*, the fusion cassette MoAlo1-GFP was introduced to the *MoALO1* null mutant. As shown in Figure 1C, the green fluorescence of MoAlo1-GFP was found in conidia and the germ tube. To confirm whether MoAlo1-GFP localized to the mitochondria, we labeled the mitochondria with Mitotracker, a red fluorescent mitochondrial dye. The green fluorescence of MoAlo1-GFP was found to overlap with the mitochondria stained with Mitotracker (Figure 1C). These data indicate that MoAlo1 is localized to the mitochondria in *M*. *oryzae*.

### 3.2. MoAlo1 Is Important for Vegetative Growth and Conidiogenesis

In order to verify the biological function of *MoALO1*, we constructed a gene-deletion cassette to knock out the target gene *MoALO1* as described in the Materials and Methods section. Subsequently, the transformants were screened by a double PCR protocol using the *β-tubulin* gene as a positive control as previously reported [22]. As shown in Appendix A, in addition to the band of the *β-tubulin* gene (~1 kb), Guy11 showed a characteristic band (~0.5 kb, using the primers MoALO1-S-F/MoALO1-S-R indicating the target gene). Then, the true null mutant showed a ~2.5 kb characteristic band (using the primers MoALO1-L-F and BAR-F, indicating the target gene-deletion cassette) while Guy11 did not. Furthermore, we re-confirmed that there was only one copy in the mutant by qRT-PCR, in which the *β-tubulin* gene was used as a control. Finally, the mutant containing a single copy of the gene-deletion cassette was considered the null mutant Δ*Moalo1.* Additionally, we generated the complementary strain MoAlo1C, which was complemented by reintroducing a full-length *MoALO1* gene to the Δ*Moalo1* mutant to confirm the biological roles of MoAlo1 and verify that the defects in the Δ*Moalo1* mutant were due to the deletion of *MoALO1*.

The wild-type Guy11 (WT), the Δ*Moalo1* mutant, and the complementation strain MoAlo1C were cultured on CM, YG, V8, and OMA medium (Figure 2A). The diameter of the Δ*Moalo1* mutant was 4.88 ± 0.03 cm, while the diameter of the wild type was 5.33 ± 0.08 cm (*p* < 0.01). In addition, we assessed the conidiation of the strains cultured on CM for 8 days. Compared with the wild-type strain Guy11, the Δ*Moalo1* mutant produced approximately 8.33% of the conidia produced by the wild type (Figure 2B). These results indicate that the conidiation was significantly reduced in Δ*Moalo1* (*p* < 0.01). The conidiophore formation of the mutant was subsequently analyzed. The Δ*Moalo1* mutant produced few conidiophores at 24 h post-conidial induction (hpi), while Guy11 and MoAlo1C developed multiple conidiophores at 24 hpi (Figure 2C). Subsequently, to investigate the glycogen distribution of Δ*Moalo1*, we employed potassium iodide to stain glycogen during appressorium development. There were no differences in the conidia and appressoria morphology developed by Guy11, Δ*Moalo1*, and MoAlo1C. From 0 to 24 h in the appressorium development on a hydrophobic surface, there was no significant difference in the cellular distribution of glycogen in the conidia and appressoria (Figure 2D). These data indicate that MoAlo1 plays key roles in growth and conidiogenesis.

### 3.3. MoAlo1 Plays a Role in Oxidative Stress

D-Erythroascorbic acid is an important antioxidant molecule in *S. cerevisiae* [15]. To examine the role of MoAlo1 in oxidative stress, we monitored the effects of oxidants on the Δ*MoAlo1* mutant. Mycelial growth was measured on YG plates supplemented with H_2_O_2_. In mycelial growth assays, the Δ*Moalo1* mutant showed significantly increased sensitivity towards H_2_O_2_ compared with the wild-type strain Guy11 and the complementary strain MoAlo1C (Figure 3), indicating that MoAlo1 plays an important role in oxidative stress.

### 3.4. MoAlo1 Is Required for Pathogenicity

To comprehensively evaluate the pathogenicity of MoAlo1 in *M. oryzae*, we performed assays on two susceptible hosts (barley and rice) to determine the pathogenicity of Δ*Moalo1.* Eight-day-old barley leaves were inoculated with mycelial agar plugs of Guy11, Δ*Moalo1*, and MoAlo1C grown on CM and photographed at 4 dpi. Typical severe disease symptoms caused by MoAlo1C were comparable to those of wild-type Guy11, but the Δ*Moalo1* mutant showed severely reduced pathogenicity. No visible lesion was observed on the barley leaves inoculated by mycelial agar plugs of Δ*Moalo1* (Figure 4A). In order to further determine the pathogenicity of the Δ*Moalo1* mutant, we designed the concentration of the conidial suspension to have a gradient and carried out a pathogenicity test on rice leaves to further explore the effect of MoAlo1 on disease progression (Figure 4B). Interestingly, both Guy11 and Δ*Moalo1* can cause severe lesions; however, with the decrease in the concentration gradient of the conidial suspension, the Δ*Moalo1* mutant showed more limited and smaller lesions than those in the wild type. Similarly, in spray infection assays with 2-week-old rice seedlings, numerous typical spindle-shaped lesions were observed on leaves sprayed with Guy11 or the complementary strain MoAlo1C, but slightly limited necrosis was observed in the Δ*Moalo1* mutant at 7 dpi (Figure 4C). The areas of the diseased lesions in 5-cm-long infected leaves caused by Δ*Moalo1* were significantly smaller than those caused by the wild-type Guy11 and the complementary strain MoAlo1C at 7 dpi (Figure 4D). Then, exogenous EASC was added to the conidial suspension and a pathogenicity analysis was carried out. As shown by Figure 4E, when exogenous EASC was added to the conidial suspension, the defective pathogenicity of the Δ*Moalo1* mutant was restored. Thus, MoAlo1 was observed to play an important role in pathogenicity.

### 3.5. MoAlo1 Play Pleiotropic Roles in Penetration

Due to the penetration difference between conidia and hyphae, we tested the penetration of the appressoria developed by conidia and the appressorium-like structures (ALS) developed by the hyphal tips. As shown in Figure 5A, there were no significant differences between the appressorial penetration rate of Guy11 and Δ*Moalo1*. Interestingly, ALS formed by the hyphal tips showed significant differences between Guy11 and Δ*Moalo1*. At 2 dpi, Guy11 ALS formed invasive hyphae that extended from cell to cell. At 4 dpi, new ALS were produced by the invasive hyphal tips of Guy11 (Figure 5B). In parallel assays, the Δ*Moalo1* hyphal tips formed ALS at 2 dpi. However, they could not form invasive hyphae to penetrate the barley leaves, although the assays were extended to 6 dpi (Figure 5B).

### 3.6. The ALO Domain and FAD_Binding_4 Domain Can Restore the Defects in the MoALO1 Null Mutant

Pfam domain analysis of MoAlo1 showed that MoAlo1 has two typical domains: the ALO domain (PF04030) and the FAD_binding_4 (PF01565). To further investigate the functions of these two domains, the ALO domain or the FAD_binding_4 domain was complemented with Δ*Moalo1*, respectively. To determine the role of Δ*Moalo1*::ALO and Δ*Moalo1*::FDA on vegetative hyphal growth, both of these strains were incubated on CM, YG, A8, and OMA medium (Figure 2A). At 8 dpi, Δ*Moalo1*::ALO showed a comparable growth rate to that of the wild type Guy11; the diameter of the Δ*Moalo1*::ALO was 5.32 ± 0.04 cm, while that of the wild type was 5.33 ± 0.08 cm. However, Δ*Moalo1*::FDA showed a comparable growth rate to that of Δ*Moalo1;* the diameter was 4.75 ± 0.05 cm for Δ*Moalo1*::FDA, while the diameter was 4.88 ± 0.03 cm for Δ*Moalo1.* These results indicate that the ALO domain is required for fungal growth. In addition, to verify if the ALO domain or the FAD_binding_4 domain is required for pathogenicity, we carried out a spray infection assay of these two strains in rice. The disease symptoms caused by both Δ*Moalo1*::FDA and Δ*Moalo1*::ALO showed severe typical lesions bearing similarity to the wild-type strain (Figure 4C,D). In general, re-introduction of either domain can restore the pathogenicity of Δ*Moalo1*, suggesting that both the ALO domain and the FAD_binding_4 domain are necessary for fungus development and pathogenicity.

## 4. Discussion

Ascorbic acid (L-ascorbic acid, ASC) is an effective antioxidant molecule and plays an indispensable role in biological antioxidants [8,9,10,27]. However, there are many reports suggesting that ASC is rare or even absent in microorganisms, but a five-carbon analog, D-erythroascorbate (EASC), whose physicochemical properties are similar to those of ASC, may play an important role in microorganisms as a substitute for ASC [11]. However, EASC has not been well-studied to date. In this study, we preliminarily identified a protein, MoAlo1, that is involved in the final step of the D-erythroascorbic acid biosynthesis pathway. In addition, we investigated its roles in vegetative growth, conidiation, and pathogenicity in the phytopathogenic fungus *M. oryzae*.

In *S. cerevisiae*, ALO catalyzes the final step of the synthesis of EASC, but when L-galactose or L-galactono-lactone is provided as the substrate, ASC can be also generated [28]. However, in *S. cerevisiae* and *T. brucei*, it exhibits a higher affinity for D-arabinono-γ-lactone; thus, EASC is the main form of ASC in *S. cerevisiae* and *T. brucei* [29]. In *Phycomyces blakesleeanus* [19], EASC was detected to be active during the growth phase, but decreased in the stationary phase, illustrating that EASC is relevant to fungi growth [16] and may play different roles in different stages during fungi growth. Glycosylation of ascorbic acid analogues seems to be a common feature in fungi [19]. EASC can be glycosylated to D-erythroascorbate glucoside, which exhibits high stability in oxidizing conditions [19]. D-erythroascorbate glucoside can be stored in non-growing cells, such as sclerotia [19] and conidia [30], which matter for the storage and metabolism of organic materials. In general, EASC is of great significance to the execution of the normal functions of organisms.

Due to the different substrates, the synthesis of ASC is different in different organisms. However, among animals, plants, and fungi, the similar enzyme proteins contain both the FAD_binding_4 domain and the ALO domain (Figure 1A) and show a similar evolutionary relationship to these organisms (Figure 1B), indicating that Alo1 is conserved in different organisms. In general, according to the characteristics of different organisms, the location of the enzyme that synthesizes ASC in different organisms is also different. Plant GALDH and yeast ALO are mitochondrial enzymes, while in mammals, GLO is located in the endoplasmic reticulum [13,17,31]. Similar to *S*. *cerevisiae*, MoAlo1 is located in the mitochondria (Figure 1C), suggesting that MoAlo1 may function as *ScAlo1*. Disruption of some mitochondrial proteins showed significant conidiogenesis and germination defects [32], consistent with Δ*Moalo1.* Mitochondria are some of the most important organelles of cells. In plants, the biosynthesis of ASC in mitochondria is related to the electron transport chain between complexes III and IV. Cytochrome c, which delivers electrons between complexes III and IV, seems to be the specific electron acceptor of GalL dehydrogenase, which catalyzes the synthesis of ASC in plants [33]. In *C. albicans* [18], a lack of *ALO* may inhibit the electron transfer of cytochromes, thereby activating cyanide-resistant respiration (CRR), which is an alternative respiratory mode conferred by alternative oxidase (AOX) [31] and leads to the accumulation of hydrogen peroxide in *C. albicans.* Because of MoAlo1’s role in blocking the electron transport chain, we speculated that MoAlo1 can be used as a blocker of electron transfer between complexes III and IV, but these pathways need to be further examined to determine whether various aspects of ASC biochemistry in these important pathogens have potential as drug targets.

As previously reported, a deficiency in *ALO* can cause physiological defects, such as stunted growth, morphological changes, and attenuated virulence [16,34]. Studies have shown that ASC may play a role in the initiation of cell proliferation in the early stage [29]. In *M. oryzae*, deletion of *MoALO1* resulted in a decrease in growth on different media and exhibited lower growth rates compared with Guy11 (Figure 2A). In addition, the Δ*Moalo1* mutant produced a striking defect in conidiogenesis (Figure 2B,C) and pathogenicity (Figure 4). In the field, *M. oryzae* spreads through conidia drifting with the wind and rain; thus, conidiogenesis is of great significance to the infection cycle and disease dissemination during epidemics [35]. Targeted deletion of many genes is accompanied by conidia defects and attenuated virulence [36,37,38,39,40], which indicates that blocking conidia may be an effective disease control strategy [41], and MoAlo1 is likely to be of great value in field control due to the defect in conidiogenesis in Δ*Moalo1*. On a hydrophobic surface or the surface of the host plant, the rice blast fungus *M. oryzae* can form appressorium-like structures (ALS) that are similar to the appressoria developed by conidia. Figure 5 shows that the Δ*Moalo1* mutant blocked ALS penetration in barley, indicating that the loss of pathogenicity in the mycelial agar test was due to the defect in ALS infection, and MoAlo1 may play pleiotropic roles in penetration. However, the Δ*Moalo1* mutant displayed a normal glycogen distribution as well as conidia morphology (Figure 2D). When exogenous EASC was added to the conidia suspension, the Δ*Moalo1* mutant showed a comparable pathogenicity to that of the wild-type Guy11, indicating that the defective pathogenicity caused by the deletion of *MoALO1* may be due to the decrease in EASC in *M. oryzae*. These results indicate that MoAlo1 is involved in the pathogenicity process of conidia and hyphae, but it seems to have little effect on the morphology. Re-introduction of any domains of ALO or the FAD_binding_4 domain showed comparable pathogenicity to that of Guy11, which indicates that both domains are required for pathogenicity.

ASC is an important antioxidant, although the antioxidant function has been extensively studied in plants and other organisms [42], there are few clear answers about the precise function of ASC in fungi. Moreover, the exact function of its analog EASC also remains a mystery. Previous reports suggest that EASC may play significant roles in the synthesis of certain substances, such as oxalate [43]. In the normal physiological function of cells, there are some defense mechanisms against reactive oxygen species (ROS), and Alo1 is an antioxidant enzyme triggered by ROS [16]. It has been reported that Alo1 participates in the antioxidant defense against parasites [34] as well as some fungi [15,16]. In addition, overexpression of Alo1 leads to the tolerance of paraquat as well as aluminum toxicity in *S*. *cerevisiae* [44]. In *M*. *oryzae*, compared with Guy11, Δ*Moalo1* was found to be more sensitive to exogenous H_2_O_2_ (Figure 3). These results indicate that MoAlo1 may be an antioxidant molecule that plays an important role in normal cell function, similarly to ScAlo1 [15].

In general, we analyzed the biological functions of MoAlo1 in *M. oryzae*. MoAlo1 localized in mitochondria and played important roles in growth, conidiogenesis, and phytopathogenicity. Moreover, MoAlo1 plays an important role in oxidation stress. However, the specific biological functions of Alo1 in *M. oryzae* still need to be further explored.

## Figures and Tables

**Figure 1 jof-08-00072-f001:**
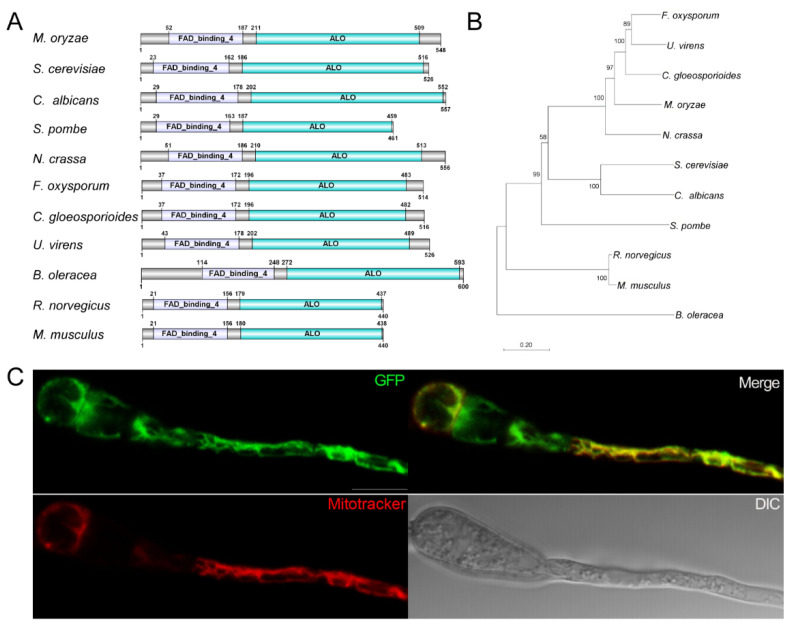
Conserved domains of ALO-containing proteins and the location analysis of MoAlo1. (**A**) Schematic diagram of ALO-containing proteins in different organisms. The SMART program (http://smart.embl-heidelberg.de, accessed on 1 November 2021) was used to perform the sequence analysis. Among filamentous fungi, plants, and mammals, ALO and FAD_binding_4 domains are conserved. Genbank number: *Saccharomyces cerevisiae* (NP_013624.1), *Candida albicans* (KGQ99802.1), *Schizosaccharomyces pombe* (NP_593526.1), *Neurospora crassa* (XP_965288.1), *Fusarium oxysporum* (KAF5266425.1), *Colletotrichum gloeosporioides* (KAF3807304.1), *Ustilaginoidea virens* (XP_042995658.1), *Brassica oleracea* (XP_013629960.1), *Rattus norvegicus* (NP_071556.2), and *Mus musculus* (NP_848862.1). (**B**) Phylogenetic analysis and classification of MoAlo1 and homologues proteins in different organisms. The phylogenetic tree was established with full sequences using the Neighbor-Joining method with 1000 bootstrap replications. (**C**) The subcellular localization of MoAlo1. The conidia and germ tube were stained with 100 nM MitoTracker^®^ Red CMXRos (Invitrogen, Carlsbad, CA, USA) and observed under a fluorescence microscope. Bar = 10 μm.

**Figure 2 jof-08-00072-f002:**
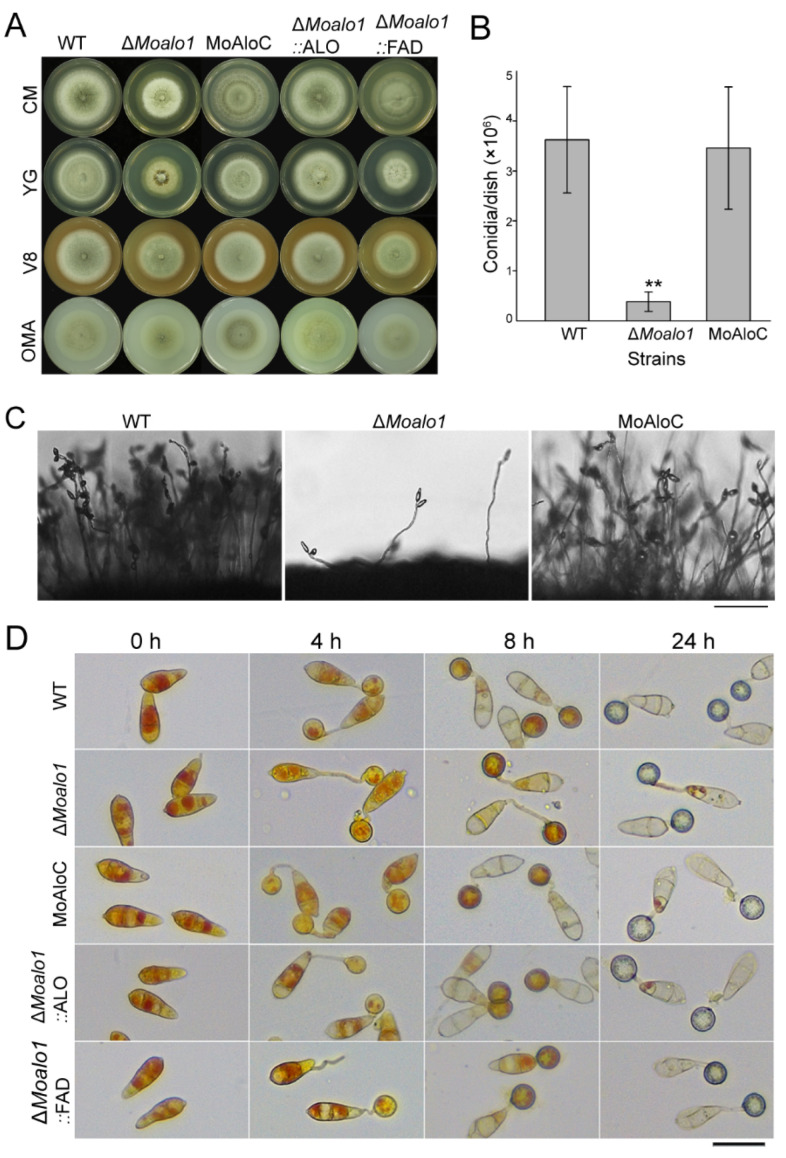
Deletion of *MoALO1* influences mycelial growth and conidiogenesis of *M*. *oryzae*. (**A**) Colony morphology of Guy11, Δ*Moalo1*, the complementary strain MoAlo1C, Δ*Moalo1*::ALO, and Δ*Moalo1*::FDA. Strains were cultured on CM, YG, V8, and OMA plates at 25 °C for 8 days before photography. (**B**) The number of conidia produced by Guy11 and Δ*Moalo1*. Prism 7.0 software was used to analyze the data. Asterisks indicate statistically significant differences (*t* test, ** *p* < 0.01). (**C**) Microscopic observation of asexual development. Conidiophores and conidia were observed after 24 h of constant light induction. Bar = 100 μm. (**D**) Glycogen distribution of the conidia and appressoria of strains. Glycogen was stained with KI/I_2_ solution and appears as dark brown deposits in conidia and appressoria. Bar = 30 μm.

**Figure 3 jof-08-00072-f003:**
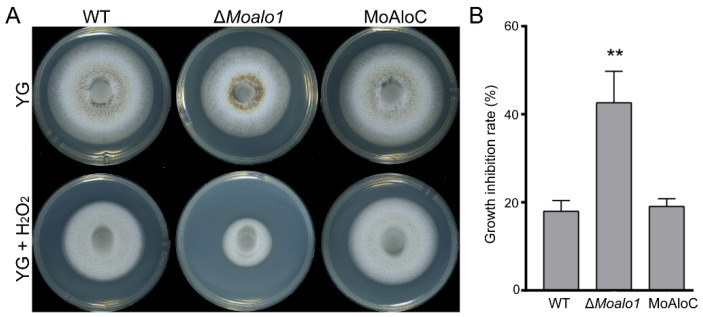
MoAlo1 plays a role in oxidative stress. (**A**) Colony morphology of Guy11, Δ*Moalo1*, and the complementary strain MoAlo1C. Strains were cultured on YG and YG plates supplemented with H_2_O_2_ (10 mM), then cultured at 25 °C for 8 days before photography. (**B**) The growth inhibition rate of the strains. Prism 7.0 software was used to analyze the data. Asterisks indicate statistically significant differences (*t* test, ** *p* < 0.01).

**Figure 4 jof-08-00072-f004:**
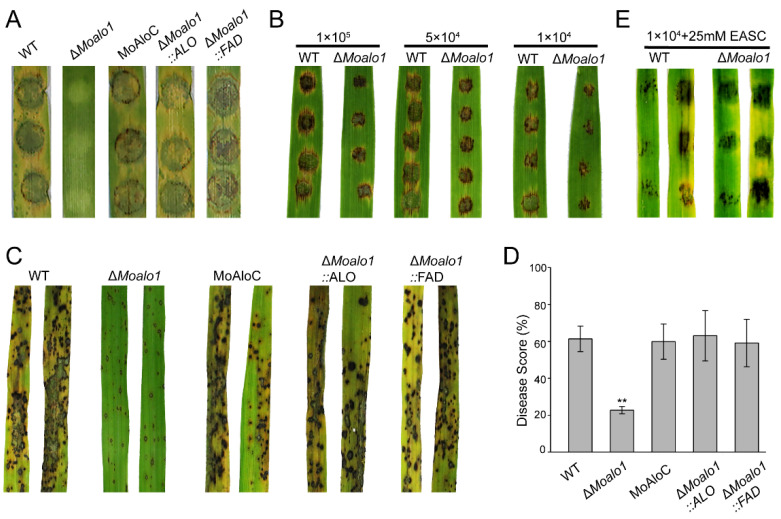
Pathogenicity assays. (**A**) Pathogenicity on barley leaves. Mycelial agar plugs were inoculated on 7-day-old barley leaves and photographed at 4 dpi. (**B**) Pathogenicity on rice leaves. Twenty microliters of conidial suspension with different concentrations (1 × 10^5^, 5 × 10^4^, and 1 × 10^4^ conidia/mL) were inoculated on 2-week-old rice leaves and photographed at 5 dpi. (**C**) Pathogenicity on rice seedlings. Conidial suspensions (1 × 10^4^ conidia/mL) of Guy11, Δ*Moalo1*, and the complementary strain MoAlo1C and mutant re-introduction of the FAD_binding_4 domain or the ALO domain were inoculated on 2-week-old rice seedlings and photographed at 7 dpi. (**D**) The disease score of the strains. After inoculation for 7 days, the rice seedlings were photographed and then the pathogenicity was calculated via PS software by calculating the area of the diseased spot. The disease score = the area of the diseased spot/the area of the total leaf. Prism 7.0 software was used to analyze the data. Asterisks indicate statistically significant differences (*t* test, ** *p* < 0.01). (**E**) Pathogenicity test upon adding exogenous EASC. EASC (25 mM) was added to a 1 × 10^4^ conidia/mL conidial suspension, and 20 μL of the conidial suspension with EASC added was dropped onto detached rice leaves and photographed at 4 dpi.

**Figure 5 jof-08-00072-f005:**
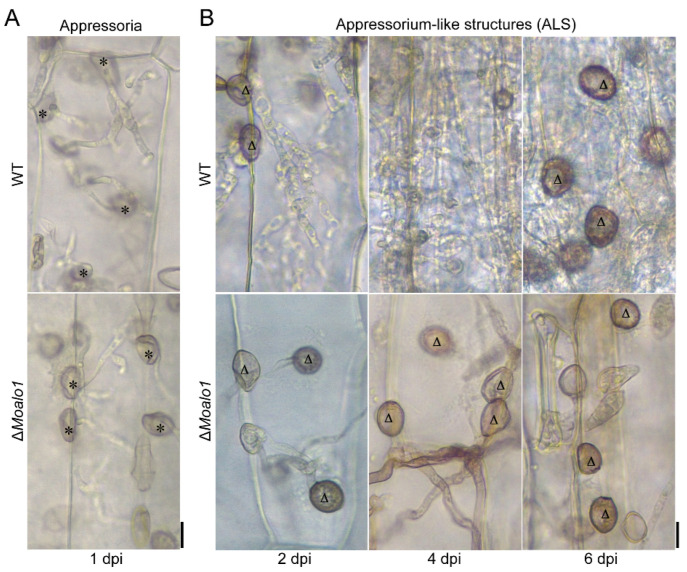
The penetration test of Δ*Moalo1* in barley. (**A**) The penetration test of the appressoria developed by conidia. Photographed after inoculation on cut barley leaves for 1 day. Asterisks indicate appressoria (**B**) The penetration test of the appressorium-like structures (ALS). Photographed at 2, 4, and 6 dpi on barley leaves. Triangles indicate ALS. Bar = 10 μm.

## Data Availability

The data are contained within the article or the Appendix A.

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
