# Peer review of "A Putative D-Arabinono-1,4-lactone Oxidase, MoAlo1, Is Required for Fungal Growth, Conidiogenesis, and Pathogenicity in Magnaporthe oryzae"

_jof, 2022, doi:10.3390/jof8010072_

Round 1
Reviewer 1 Report
The author has identified MoAlo1 gene containing ALO and FAD_binding_4 domain which is supposed to be involved in the final step of D-erythroascorbic acid biosynthesis pathway in Magnaporthe oryzae Guy11 as one of the homologue of Saccharomyces cerevisiae ALO1. MoAlo1 localisation studies have been done in M. oryzae using GFP tagged MoAlo1 and found to be localized in the mitochondrial region. Total gene deletion strategy has been applied to generate ΔMoAlo1 null mutant and found to be defective in sporulation, conidiophores number and arrangement, growth inhibition and ultimately leads to less virulent on rice as well as on barley leaf. ΔMoAlo1 null mutant and found to be sensitive to oxidative stress. Pleiotropic effect of ΔMoAlo1 null mutant leads to defective penetration of appressoria. Phenotypic effect of the knock out mutant could be reverted by reintroduction of the full length MoAlo1C as well as FAD_binding_4 and ALO domains individually into ΔMoAlo1 null mutant
Major comments: The putative MoAlo1 was identified by sequence homology only. But it is not clear whether the protein encoded by this gene indeed have desired biochemical function. There is no direct biochemical evidence with the purified MoAlo1. Authors also did not check the intracellular level of D-erythroascorbic acid (EASC) in ΔMoAlo1 null mutant. Does MoAlo1 complement S cerevisiae mutant? More intriguing is the phenotype re-version of null mutant by individual FAD or Alo1 domain. Does this individual domain possess the desired enzymatic activity? Without these answers the manuscript looks very preliminary and incomplete.
Minor Comments:
Details of the cloning strategy, constructs and selection is missing.
Line 74: ΔMoalo1::FAD not ΔMoalo1::FDA
Line 113: using sterile water
Line 125: 4 days of inoculation
Figure 2, Line 212-214: ΔMoalo1 should be written everywhere instead of Moalo1 (Figure 2D). How did authors induce appressoria formation and staining? Did authors also checked for cellulose, lignocellulose and chitin in the mutant (ΔMoalo1) and wild type?
Line 218-219: Moalo1 instead of MoAlo1
Line 220: increased sensitivity towards H2O2
Line 226: Please mention the concentration of H2O2 used here in legend
Author Response
Review Report 1
Comments and Suggestions for Authors
The author has identified MoAlo1 gene containing ALO and FAD_binding_4 domain which is supposed to be involved in the final step of D-erythroascorbic acid biosynthesis pathway in Magnaporthe oryzae Guy11 as one of the homologue of Saccharomyces cerevisiae ALO1. MoAlo1 localisation studies have been done in M. oryzae using GFP tagged MoAlo1 and found to be localized in the mitochondrial region. Total gene deletion strategy has been applied to generate ΔMoAlo1 null mutant and found to be defective in sporulation, conidiophores number and arrangement, growth inhibition and ultimately leads to less virulent on rice as well as on barley leaf. ΔMoAlo1 null mutant and found to be sensitive to oxidative stress. Pleiotropic effect of ΔMoAlo1 null mutant leads to defective penetration of appressoria. Phenotypic effect of the knock out mutant could be reverted by reintroduction of the full length MoAlo1C as well as FAD_binding_4 and ALO domains individually into ΔMoAlo1 null mutant
Major comments: The putative MoAlo1 was identified by sequence homology only. But it is not clear whether the protein encoded by this gene indeed have desired biochemical function. There is no direct biochemical evidence with the purified MoAlo1. Authors also did not check the intracellular level of D-erythroascorbic acid (EASC) in ΔMoAlo1 null mutant. Does MoAlo1 complement S cerevisiae mutant? More intriguing is the phenotype re-version of null mutant by individual FAD or Alo1 domain. Does this individual domain possess the desired enzymatic activity? Without these answers the manuscript looks very preliminary and incomplete.
Response: As for the identified of MoAlo1, we have conducted a bioinformatics analysis of MoAlo1 through domains structural analysis and phylogenetic tree analysis, then the functions of MoAlo1 were further confirmed by knocking out the gene. In order to explore the intracellular level of D-erythroascorbic acid (EASC) in ΔMoalo1 null mutant, we have tried to detect the content of EASC using HPLC which described in Saccharomyces cerevisiae [1] and Candida albicans [2]. But we did not get exact results. But, ΔMoalo1 null mutant can restore the pathogenicity when exogenous EASC were added (Newly added in the article, Figure 4E). Therefore, we believe that the defected in pathogenicity of the ΔMoalo1 null mutant may be caused by the lack of EASC content. As for “Does MoAlo1 complement S cerevisiae mutant?” Due to limited conditions, we are very sorry that we can’t achieve it. Moreover, the deletion of either domains of FAD biding _4 or ALO domain does not affect its pathogenicity, so we speculated that pathogenicity and EASC requires both domains to function at the same time. To sum up, the pathogenic defect was rescued after adding exogenous EASC, indicating that the main reason for the pathogenic defect of ΔMoalo1 null mutant is the lack of EASC.
Here are the references:
- Huh, W.K.; Lee, B.H.; Kim, S.T.; Kim, Y.R.; Rhie, G.E.; Baek, Y.W.; Hwang, C.S.; Lee, J.S.; Kang, S.O. D-Erythroascorbic acid is an important antioxidant molecule in Saccharomyces cerevisiae. Mol Microbiol 1998, 30, 895-903, doi:10.1046/j.1365-2958.1998.01133.x.
- Huh, W.K.; Kim, S.T.; Yang, K.S.; Seok, Y.J.; Hah, Y.C.; Kang, S.O. Characterisation of D-arabinono-1,4-lactone oxidase from Candida albicans ATCC 10231. Eur J Biochem 1994, 225, 1073-1079, doi:10.1111/j.1432-1033.1994.1073b.x.
Furthermore, we revised the manuscript in accordance with the reviewer’s comments, and carefully proof-read the manuscript to minimize typographical, grammatical, and bibliographical errors.
Here below is our description on revision according to the reviewer’s comments.
Minor Comments:
- Details of the cloning strategy, constructs and selection is missing.
Response: Thanks for the reviewer’s good suggestion, we have added in the article.
- Line 113: using sterile water
Response: Thanks for pointing out the mistake, we have revised in the article.
- Line 125: 4 days of inoculation
Response: Thanks for pointing out the mistake, we have revised in the article.
- Figure 2, Line 212-214: ΔMoalo1 should be written everywhere instead of Moalo1 (Figure 2D). How did authors induce appressoria formation and staining? Did authors also checked for cellulose, lignocellulose and chitin in the mutant (ΔMoalo1) and wild type?
Response: Thanks for pointing out the mistake, we have revised in the article. For the distribution of glycogen experiment, conidia were harvested and dilute to 5×104 conidia / mL, then 20uL conidia suspension were dropped on the plastic film to induce the appressorium formation, then cultivated in the dark for 0, 4, 8, 24 hours. Observed under the microscope after stained with potassium iodide for 1 min. We have added it in the article.
The distribution of glycogen experiment was carried out due to the infection of M. oryzae was caused by the mechanism force develop from the accumulation of cargo such as glycogen in appressorium. It would be an interesting research to check for cellulose, lignocellulose and chitin in the mutant (ΔMoalo1) and wild type. However, it is not the scope of this current work, we would love to investigate it in the future.
Line 218-219: Moalo1 instead of MoAlo1
Response: Thanks for the reviewer’s suggestion. The naming of Magnaporthe oryzae follows the rules: In Magnaporthe oryzae, “Mo’” stands for Magnaporthe oryzae. The first letter of protein should be capitalized, eg: MoAlo1. All the letters of gene should be capitalized and italicized, eg: MoALO1. The letters should be lowercased and italicized in mutant, eg: ΔMoalo1. So here we used “MoAlo1”.
- Line 226: Please mention the concentration of H2O2 used here in legend
Response: Thanks for the reviewer’s good suggestion, we have added in the article.

Reviewer 2 Report
Dear Authors,
Your article is very interesting and will certainly be of interest to readers, but section "Materials and methods" is no good at all! It is absolutely irreproducible according to your meager description, there are no references to the original methods in the text. You would to publish an experimental article, so it should contain a clear description of the experiments.
Please correct this point, this is the main comment on your article.
And some small comments:
- Line 20: Decrypt "ALO"
- Line 36: Do you mean the SI derived unit of pressure MPa?
- Line 73: What is the sourse of fungal strains?
- Line 142: Decrypt "Pfam"
- Figure 1 (A) is very small, I do not see anything!
Author Response
Thank you for your letter and the reviewer’s comment concerning our manuscript. Those comments are valuable and very helpful. We have read through comments carefully and have made corrections, especially in "Materials and methods".
Here below is our description on revision according to the reviewer’s comments.
Comments and Suggestions for Authors
Dear Authors,
Your article is very interesting and will certainly be of interest to readers, but section "Materials and methods" is no good at all! It is absolutely irreproducible according to your meager description, there are no references to the original methods in the text. You would to publish an experimental article, so it should contain a clear description of the experiments.
Please correct this point, this is the main comment on your article.
And some small comments:
- Line 20: Decrypt "ALO"
Response: Thanks for the reviewer’s good suggestion, ALO means: D-arabinono-1,4-lactone oxidase, we have added in the article.
- Line 36: Do you mean the SI derived unit of pressure MPa?
Response: Thanks for the reviewer’s good suggestion. Yes, it is the SI derived unit of pressure MPa. We have changed the MPa to 106 Pa in the article.
- Line 73: What is the source of fungal strains?
Response: Thanks a lot for pointing out that. We have bought the fungal strains of wild type Guy11 from ATCC: The Global Bioresource Center.
- Line 142: Decrypt "Pfam"
Response: Thanks for the reviewer’s good suggestion. Pfam is a database of protein families and domains that is widely used to analyze novel genomes, metagenomes and to guide experimental work on particular proteins and systems.
- Figure 1 (A) is very small, I do not see anything!
Response: Thanks for the reviewer’s good suggestion. We have changed the Figure 1 (A) in the article.
Reviewer 3 Report
I hope that you find the attached points valuable and easy to follow and then be constructive for your manuscript. You have taken good initiative in documenting these criteria to the rice blast fungus. However, some need to be better weighed and measured.
- Title,
MoAlo1, a putative D-arabinono-1,4-lactone oxidase, is re- 2 quired for fungal development and pathogenicity in Magnaporthe oryzae
- It is not usual starting the title with Acronym or abbreviation
- Fungal development is very general need to more specific to the work.
- Line 46..it (is) located..
- Line 50-51 not clear, meaning what..
- AsC / not found in Ref # 11, 17 and 18
- Should include a list of Acronyms / abbreviation used/ introduced here. Should consolidate all in the same way presented. All in one font all in one form not as it is here same one in all caps or mixed fonts or in italic. Full name should precede the acronym.
- YG medium not defined what kind of fungal media
- Line 83.. tense !!
- Line 93.. confusing names and abbreviation
- Line 114.. nonscientific PHRASE of experimental description
- Line 121.. not correct phrase. A host is an organism (eg.: a plant) that is harboring a parasite or pathogen from which it obtains its nutrients. The host range refers to the various kinds of host plants that a given pathogen may parasitize. A host is considered resistant when it has the ability to exclude, hinder or overcome the effects of a given pathogen or other damaging factor. A plant may be resistant to one pathogen or condition but not others. Tolerance is the ability of a plant to be colonized by a pathogen or exposed to an abiotic factor without dying or demonstrating disease symptoms. Susceptibility is the antithesis of resistance.
- Line 116.. drug; is not suitable term. How did you weight the magnitude of inhibition??
- Line 124..Not clear how this; Virulence magnitude figured??
- . with a hemocytometer. using …
- Line 127.. dropped on rice seedling.. or place on leaves of…
- Line 128..dpi should be written in full first..
- Line 133..wrong phytopathogenic description. Pathogenicity is not for the appressorium or the infection beg. And the latter one is of the germ tube.
- Line 135.. what is ALS
- Lines 137-138..what are the rules of making these abbreviation and how is it followed out??
- Line 142..Pfam..??
- Line 143 .. CLC..?? CLC Genomics Workbench (QIAGEN)..???
- Lines 142-144.. check meaning? Not clear!!
- Line 145 .. BlastP define / clarify.
- Line 154.. change “showed” to Showen
- Line 158.. is it “hypha” or “germ tube”??
- Line 185.. mutant and complementation strain (MoAlo1C), need not be in parenthesis.
- Line 187.. “slightly” and Line 190 “fewer”.. not exact scientific description. A difference of 0.08cm is not significant. Furthermore, the figure A does not show that.
- Line 209..not “error bar”, it range or deviation around a mean value..
- Lines 219-220.. significantly increased sensitivity of the H2O2 compared to the wild-type strain Guy11 and complementary strain MoAlo1C (Figure 3). How did you measure and quinate the SENSITIVITY??? And convert to % ? What is 100% ??
- Line 226.. and everywhere. Whenever RATE is mentioned, it means change along time!!. You used “rate” in a nonscientific way and in a very ambiguous manner. It should be adjusted.
How did you measure Growth rate and inhibition rate and sensitivity rate?
- Line 226.. how much peroxide need to be here too.
- Line 227.. again Growth rate
- Line 236..clarify is it Virulence or Infection??
- Line 259.. Disease score, in text and on the graph Fig 4. Not mentioned, not clear how was that calculated, figured and find out?? Score on what bases and comparison???.
Also, how much is 4 X 10 to power 4 compared to other concentrations. Whay not describe, explained in the Material and methods section??
- Fig 4 D, Disease score in % what is 100%??
- Lines 268-271.. Not clear on figures.. need pointers posted on the figures showing or indicating what is meant by that description.
- Line 272.. this is not supported by findings here.
- Line 287 Growth rate .. again .. should be adjusted to what the way or manner measured and weighed’
- Line 289 .. Not supported for growth by findings..
- Line 290.. Not supported for Virulence by findings..
Discussion
- Too much of a literature rehash rather than a discussion
- Lines 312-317 to 349 .. away from the present discussion of results.
- Not exact in putting these finding in perspective position to the findings of other research.
Some of the most recent literature not reviewed such as for example;
D-Erythroascorbic acid is an important antioxidant molecule in Saccharomyces cerevisiae
Won-Ki Huh,Byung-Hoon Lee,Seong-Tae Kim,Yeon-Ran Kim,Gi-Eun Rhie,Yong-Woon Baek … See all authors
First published: 01 May 2002
https://doi.org/10.1046/j.1365-2958.1998.01133.x
Citations: 79

Author Response
Review Report 3
Thank you very much for your attention and the referee’s evaluation and comments on our paper. We have revised the manuscript according to your kind advices and detailed suggestions. Here below is our description on revision according to the reviewer’s comments.
Comments and Suggestions for Authors
I hope that you find the attached points valuable and easy to follow and then be constructive for your manuscript. You have taken good initiative in documenting these criteria to the rice blast fungus. However, some need to be better weighed and measured.
Title,
- MoAlo1, a putative D-arabinono-1,4-lactone oxidase, is required for fungal development and pathogenicity in Magnaporthe oryzae
- It is not usual starting the title with Acronym or abbreviation
- Fungal development is very general need to more specific to the work.
Response: Thanks for the reviewer’s good suggestion. We have changed the title to "A putative D-arabinono-1,4-lactone oxidase, MoAlo1, is required for fungal growth, conidiogenesis and pathogenicity in Magnaporthe oryzae”.
- Line 46..it (is) located..
Response: Thanks a lot for pointing out that. We have made the correction.
- Line 50-51 not clear, meaning what..
Response: Thanks a lot for pointing out the mistakes. We have already changed it in the article.
- AsC / not found in Ref # 11, 17 and 18
Response: Thanks a lot for pointing out the mistakes. We have already changed it in the article. L-ascorbic acid (ASC) has been mistakenly written as AsC, and we have re-reviewed the Ref #11, 17 and 18, and further confirmed the correct citation of references.
- Should include a list of Acronyms / abbreviation used/ introduced here. Should consolidate all in the same way presented. All in one font all in one form not as it is here same one in all caps or mixed fonts or in italic. Full name should precede the acronym.
Response: Thanks for the reviewer’s good suggestion. The naming of Magnaporthe oryzae follows the rules: In Magnaporthe oryzae, “Mo” stands for Magnaporthe oryzae. The first letter of protein should be capitalized, eg: MoAlo1. All the letters of gene should be capitalized and italicized, eg: MoALO1. The letters should be lowercased and italicized in mutant, eg: ΔMoalo1.
- YG medium not defined what kind of fungal media
Response: Thanks for the reviewer’s good suggestion. YG medium: Yeast glucose medium (YG), were used to cultivate fungi as previously described. Furthermore, we have redefined it in the article.
Here is the reference:
Liu, X.H.; Chen, S.M.; Gao, H.; Ning, G.A.; Shi, H.B.; Wang, Y.; Dong, B.; Qi, Y.Y.; Zhang, D.M.; Lu, G.-d.; et al. The small GTPase MoYpt7 is required for membrane fusion in autophagy and pathogenicity of Magnaporthe oryzae. Environmental Microbiology 2015, 17, doi:10.1111/1462-2920.12903.
- Line 83.. tense !
Response: Thanks for the reviewer’s good suggestion. We have made the correction.
- Line 93.. confusing names and abbreviation
Response: Thanks a lot for pointing out the mistakes. We have incorrectly written the full name of the bacterial bialophos resistance gene (BAR) gene; we have already revised in the article.
- Line 114.. nonscientific PHRASE of experimental description
Response: Thanks for the reviewer’s good suggestion. We have made the corrections.
- Line 121.. not correct phrase. A host is an organism (eg.: a plant) that is harboring a parasite or pathogen from which it obtains its nutrients. The host range refers to the various kinds of host plants that a given pathogen may parasitize. A host is considered resistant when it has the ability to exclude, hinder or overcome the effects of a given pathogen or other damaging factor. A plant may be resistant to one pathogen or condition but not others. Tolerance is the ability of a plant to be colonized by a pathogen or exposed to an abiotic factor without dying or demonstrating disease symptoms. Susceptibility is the antithesis of resistance.
Response: Thanks for the reviewer’s good suggestion. We have made the corrections.
- Line 116.. drug; is not suitable term. How did you weight the magnitude of inhibition??
Response: Thanks for the reviewer’s good suggestion. We have changed the word “drug” to “stress” in the article. As for how we weight the magnitude of inhibition, the diameters of the strains were measured and analyzed. The inhibition rate = (The diameter of strains growth on YG medium in 8 days - The diameter of strains growth on the YG medium supplementary with H2O2 in 8 days) / The diameter of strains growth on the YG medium in 8 days.
- Line 124..Not clear how this; Virulence magnitude figured??
Response: Thanks for the reviewer’s good suggestion. After rigorously reviewed the literature, we have changed the word “virulence” into “pathogenicity”. As for the pathogenicity of barley, it is obviously showed the difference pathogenicity between Guy11 and ΔMoalo1.
- with a hemocytometer. using …
Response: Thanks for the reviewer’s good suggestion. We have made the correction.
- Line 127.. dropped on rice seedling.. or place on leaves of…
Response: Thanks for the reviewer’s good suggestion. Rice seedlings were sprayed with a certain concentration of conidia suspension used the artist brush, and after 7 dpi, the leaves were cut, photographed and calculated by PS software. We have declared clearly in the article.
- Line 128..dpi should be written in full first..
Response: Thanks for the reviewer’s good suggestion. We have made the correction.
- Line 133..wrong phytopathogenic description. Pathogenicity is not for the appressorium or the infection beg. And the latter one is of the germ tube.
Response: Thanks for the reviewer’s good suggestion. We have made the correction.
- Line 135.. what is ALS
Response: Thanks for the reviewer’s good suggestion. ALS: appressorium-like structures (ALS). We have made the correction in the article.
- Lines 137-138..what are the rules of making these abbreviation and how is it followed out??
Response: Thanks for the reviewer’s good suggestion. The naming of Magnaporthe oryzae follows the rules: In Magnaporthe oryzae, “Mo’” stands for Magnaporthe oryzae. The first letter of protein should be capitalized, eg: MoAlo1. All the letters of gene should be capitalized and italicized, eg: MoALO1. The letters should be lowercased and italicized in mutant, eg: ΔMoalo1.
- Line 142..Pfam..??
Response: Thanks for the reviewer’s good suggestion. Pfam is a database of protein families and domains that is widely used to analyze novel genomes, metagenomes and to guide experimental work on particular proteins and systems.
- Line 143 .. CLC..?? CLC Genomics Workbench (QIAGEN)..???
Response: Thanks for the reviewer’s good suggestion. We use the integrated module of the Pfam domain to search in CLC Genomics Workbench (Qiagen,Germany) with default parameters. And we have revised in the article.
- Lines 142-144.. check meaning? Not clear!!
Response: Thanks for the reviewer’s good suggestion. We have made the correction in the article.
- Line 145 .. BlastP define / clarify.
Response: Thanks for the reviewer’s good suggestion. BlastP means: Protein Basic Local Alignment Search Tool, we have made the correction in the article.
- Line 154.. change “showed” to Shown
Response: Thanks for the reviewer’s good suggestion. We have made the correction in the article.
- Line 158.. is it “hypha” or “germ tube”??
Response: Thanks for pointing out the mistake. We have changed the world “hypha” to “germ tube” in the article.
- Line 185.. mutant and complementation strain (MoAlo1C), need not be in parenthesis.
Response: Thanks for the reviewer’s good suggestion. We have made the correction in the article.
- Line 187.. “slightly” and Line 190 “fewer”.. not exact scientific description. A difference of 0.08cm is not significant. Furthermore, the figure A does not show that.
Response: Thanks for the reviewer’s good suggestion. We have made the correction in the article. We have changed the sentence into: “The diameter of the ΔMoalo1 null mutant was 4.88 ± 0.03 cm while the wild type was 5.33 ± 0.08 cm (P<0.001)”. 0.08cm represents the range or deviation around a mean value. Moreover, as for figure A, we have stated the relevant data in the article. Moreover, as for figure A, although its values didn’t show in the figure, we have stated the relevant data in the article.
- Line 209..not “error bar”, it range or deviation around a mean value..
Response: Thanks for the reviewer’s good suggestion. We have made the correction in the article.
- Lines 219-220.. significantly increased sensitivity of the H2O2 compared to the wild-type strain Guy11 and complementary strain MoAlo1C (Figure 3). How did you measure and quinate the SENSITIVITY??? And convert to % ? What is 100% ??
Response: Thanks for the reviewer’s good suggestion. The inhibition rate = (The diameter of strains growth on the YG medium in 8 days - The diameter of strains growth on the YG medium supplementary with H2O2 in 8 days) / The diameter of strains growth on the YG medium in 8 days. We have added in the article.
- Line 226.. and everywhere. Whenever RATE is mentioned, it means change manner. It should be adjusted.
Response: We thank the reviewer for the effort reading our manuscript and providing us with useful comments. We respectfully believe that the reviewer, but here the growth rate of the strains was defined as its growth diameter in 8 days, so we used the word “Rate” here.
- How did you measure Growth rate and inhibition rate and sensitivity rate?
Response: Thanks for the reviewer’s good suggestion. The growth rate: The growth diameter in 8 days. The inhibition rate = (The diameter of strains growth on the YG medium in 8 days - The diameter of strains growth on the YG medium supplementary with H2O2 in 8 days) / The diameter of strains growth on the YG medium in 8 days. The inhibition rate and sensitivity rate use the same calculation method.
- Line 226.. how much peroxide need to be here too.
Response: Thanks for the reviewer’s good suggestion. We have added it in the article.
- Line 227.. again Growth rate
Response: We thank the reviewer for the effort reading our manuscript and providing us with useful comments. We respectfully believe that the reviewer, but here the growth rate of the strains was defined as its growth diameter in 8 days, so we used “Rate” here.
- Line 236..clarify is it Virulence or Infection??
Response: Thanks for the reviewer’s good suggestion. We have changed the world “Virulence” into “Infection” in the article.
- Line 259.. Disease score, in text and on the graph Fig 4. Not mentioned, not clear how was that calculated, figured and find out?? Score on what bases and comparison???. Also, how much is 4 X 10 to power 4 compared to other concentrations. Why not describe, explained in the Material and methods section??
Response: Thanks for the reviewer’s good suggestion. The disease score = The area of diseased spot / The area of total leaf. 1×104 means: 1×104 conidia/mL conidial suspension. We have made the corrections in the article.
- Fig 4 D, Disease score in % what is 100%??
Response: Thanks for the reviewer’s good suggestion. The disease score = The area of diseased spot / The area of total leaf. We have added it in the article.
- Lines 268-271.. Not clear on figures.. need pointers posted on the figures showing or indicating what is meant by that description.
Response: Thanks for the reviewer’s good suggestion. We have made the correction in the article.
- Line 272.. this is not supported by findings here.
Response: Thanks for the reviewer’s good suggestion. We have made the correction in the article.
- Line 287 Growth rate .. again .. should be adjusted to what the way or manner measured and weighed
Response: We thank the reviewer for the effort reading our manuscript and providing us with useful comments. We respectfully believe that the reviewer, but here the growth rate of the strains was defined as its growth diameter in 8 days, so we used “Rate” here.
- Line 289 .. Not supported for growth by findings..
Response: Thanks for pointing out the mistake. We have changed it in the article. “After 8 dpi, ΔMoalo1::ALO showed comparable growth rate with the wild type Guy11, the diameter of the ΔMoalo1::ALO was 5.32 ± 0.04 cm while the wild type was 5.33 ± 0.08 cm. However, ΔMoalo1::FDA showed comparable growth rate with ΔMoalo1, the diameter was 4.75 ± 0.05 cm of ΔMoalo1::FDA while there was 4.88 ± 0.03 cm of ΔMoalo1. These indicating that the ALO domain is required to fungal growth.”
- Line 290.. Not supported for Virulence by findings..
Response: Thanks for the reviewer’s good suggestion. In Figure 4C, D, ΔMoalo1::ALO or ΔMoalo1::FDA showed comparable pathogenicity to that of Guy11. It is showed that knocking out any of the domains showed equivalent pathogenicity to that of the wild type, so we speculated that the pathogenicity requires these two domains to work together.
Discussion
- Too much of a literature rehash rather than a discussion
Response: Thanks for the reviewer’s good suggestion. We have rewrite the Discussion
- Lines 312-317 to 349 .. away from the present discussion of results.
Response: Thanks for the reviewer’s good suggestion. We have deleted the rdundant sentences.
- Not exact in putting these finding in perspective position to the findings of other research.
Some of the most recent literature not reviewed such as for example;
D-Erythroascorbic acid is an important antioxidant molecule in Saccharomyces cerevisiae
Won-Ki Huh,Byung-Hoon Lee,Seong-Tae Kim,Yeon-Ran Kim,Gi-Eun Rhie,Yong-Woon Baek … See all authors
First published: 01 May 2002
https://doi.org/10.1046/j.1365-2958.1998.01133.x
Citations: 79
Response: Thanks for the reviewer’s good suggestion. We have changed in the article. About the English writing of the manuscript, we have asked for native English speaker to revise the article.

Round 2
Reviewer 1 Report
The revised version is improved and addressed the major concern raised earlier. I recommend its acceptance
Reviewer 2 Report
I think this manuscript can be accepted now in present form
Reviewer 3 Report
You put good efforts in taking care of raised point. However, I suggest you get better comprehension on some of the basic trophological aspects such as Disease incidence and Disease severity.
Good you corrected growth measure per end number of days which is NOT a RATE by definition. As mentioned, Rate is change per unit time over certain time run.